



# Effect of mesoscale eddy on thermocline depth over the global ocean: deepen and uplift

Xiaoyan Chen[1], Ge Chen[1, 2]

[1]Department of Marine Technology, Institute for Advanced Ocean Study, Ocean University of China, China
[2] Qingdao National Laboratory for Marine Science and Technology, Qingdao, China

*Correspondence to*: Ge Chen (gechen@ouc.edu.cn)

**Abstract.** Existing studies on the vertical displacement of thermoclines driven by mesoscale eddies are insufficient and rare. Using 17-year Argo dataset in combination with satellite altimetry, the deepening and uplifting of the depth of thermocline (DTC) by anticyclonic (AE) or cyclonic eddies (CE), respectively, were estimated globally. DTC shifts exhibited multiple
geographic and seasonal trends, with the largest magnitude shifts occurring in March and September in the Northern and Southern Hemispheres, respectively. The more pronounced DTC shifts were concentrated in the midlatitudes, and the largest DTC displacements appeared along the western boundaries of strong current systems, with peak shifts of more than 40 m. In general, eddy-induced DTC shifts were linearly correlated with eddy radius and amplitude, suggesting that high intensity eddies induced larger DTC displacements. Finally, a normalized analysis revealed a monopole (ring) structure of DTC ringing
the eddy center inside the AE (CE). The forces of AE and CE on the DTC were different, seen in the stronger deepening at the center of the AE (~30 m) than the uplifting at the center of the CE (~20 m). One possible mechanism for this asymmetry could stem from differential current shears in the thermoclines in AE and CE.

## 1 Introduction

The thermocline separates the warm surface water from the deep cold water and is one of the most significant structural features
in marine thermodynamics. The word ''thermocline'' first appeared in the limnology literature in the late nineteenth century (Pedlosky, 2006), before which it was more frequently referred to as the discontinuity layer or transition layer (Sverdrup et al., 1942). Thermocline depth variation is closely related to vertical water movement and has also been linked to major climate phenomena such as the El Niño–Southern Oscillation (Wang et al., 2000). According to previous studies, a variety of large-scale ocean circulation and transport mechanisms can affect the thermocline. For example, low frequency wind forcing may
explain the large decadal-scale fluctuations in the thermocline observed in the center of the North Atlantic (Sturges et al., 1998). Sirven and Frankignoul (1999) found that changes in Ekman pumping created a baroclinic Rossby wave that deepened the thermocline both north and south of the subduction line and Zelle et al (2004) suggested that the thermocline depth and SST are strongly correlated which seemed to be caused primarily by the upwelling pathway and the wind coupling pathway.





However, oceanic circulation is not characterized only by large-scale currents, it is also characterized by energetic mesoscale
structures like eddies (Patrice and Lapeyre, 2009). The role eddies play in large scale ocean circulation and transport systems
has been a long-standing and observationally challenging topic. As eddies capture large volumes of water, which are in constant
motion both horizontally and vertically, they can significantly impact the horizontal and vertical distribution of local marine
substances through the physical processes of stirring and pumping (Gaube et al., 2014). Eddy pumping can induce strong
vertical exchanges of water, causing rapid changes in the vertical distributions of oceanic parameters (e.g. temperature, salinity,
potential density, nutrients, etc.; Klein and Lapeyre, 2009). Discussions about the contribution of mesoscale eddies to the
stratification and ventilation of the thermocline over the past decades have generally emphasized that eddies are dominant
drivers behind the vertical transport of thermocline waters (e.g. Roemmich and Gilson, 2001; Marshall et al., 2002; Sallee et
al., 2010). Using Argo data, Chen et al. (2011) looked at eddies in the South China Sea (SCS) and suggested that cyclonic
eddies (CE) produce shallower and thinner thermoclines, whereas anticyclonic eddies (AE) cause thermoclines to deepen and
thicken. Zu et al. (2019) pointed out that the characteristics of thermoclines associated with eddies in the SCS were primarily
determined by the balance of the horizontal and vertical advection of the background temperature and salinity fields. Recently,
some studies have concluded, using the interactions between eddies and the mixing layer as a reference, that anticyclones
deepen the mixed layer whereas cyclones thin it, and the magnitude of these eddy-induced mixed layer depth shifts were largest
in winter (Hausmann et al., 2016; Gaube et al., 2018). However, while the vertical exchange of waters between the upper and
interior ocean usually occur within or below the mixed layer, studies have shown that the wind-driven vertical velocity within
the mixed layer has little to no effect on the exchanges between the surface layers and the ocean interior (Haine and Marshall,
1998, Giordani and Caniaux, 2005). These exchanges are therefore driven mostly by forces below the mixed layer, where
tracer vertical gradients are much larger than in the mixed layer, and this vertical velocity is captured entirely by mesoscale
and submesoscale dynamics (Klein and Lapeyre, 2009; Sallee et al., 2010). These observations show that the mixed layer is
vulnerable to the influence of the local ocean environment. Thus, whether studying the vertical exchange of water or the effect
of the ocean on global climate responses, it is important to consider the thermocline and the stability of the stratification.
However, global scale studies on the vertical displacement of the thermocline driven by mesoscale eddies are insufficient and
rare.

Observational characterizations of mesoscale signatures on thermocline depth have typically been limited to single event field
campaigns and have focused more on describing this phenomenon rather than quantitative research (Chen et al., 2011; Zu et
al., 2019). This is likely because of the restrictions imposed by limited datasets on the vertical structure of the ocean in the
past. Systematic studies of thermocline depth are becoming feasible as coverage grows and the Argo profile observational
archive is maintained. With the development of marine satellite remote sensing technology, remote sensing data has been
increasingly applied to study oceanic mesoscale eddies. In this study, we exploited the Argo profile archive in combination
with the eddy dataset from our previous work on an eddy identification method to provide a systematic characterization of
eddy-induced displacements of thermocline depths in oceans worldwide.



This paper is organized as follows: In section 2 we describe the two datasets (altimetry and Argo data) used in this work. Next, in section 3, the eddy identification algorithm based on satellite data is briefly presented along with the methodology used to calculate thermocline depth and conduct normalization analysis. The results are presented in section 4. The spatio-temporal

distribution of thermocline depths inside AEs and CEs are described in section 4.1. Section 4.2 provides a calculation of eddy-induced thermocline depth shifts and relates an analysis of the influence of eddy properties on thermocline depth. In section 4.3, we describe how the spatial distributions of thermoclines in eddies were transformed from a geographical coordinate system to an eddy coordinate system using a normalization method. Additionally, a quantitative evaluation of the effect of eddies on the depth of thermoclines is presented in this section. Finally, section 5 contains a brief summary and our conclusions.

## 2 Data sets


### 2.1 Satellite altimeter data

The sea level anomaly (SLA) data used in this study were delayed time products generated by Archiving, Validation, and Interpretation of Satellite Oceanographic (AVISO) from a combination of T/P, Jason-1, Jason-2, Jason-3, and Envisat missions. The SLA dataset spanned nearly 17 years, from January 2002 to January 2019, and had a daily temporal resolution and a $(1/4)°$

$× (1/4)°$ spatial resolution.

### 2.2 Argo floats

The Argo (Array for Real-time Geostrophic Oceanography) data also spanned from January 2002 to January 2019. Argo floats observe large temporal (seasonal and longer) and spatial (thousand kilometers and larger) scale subsurface ocean variability worldwide (Roemmich et al., 2009). The Argo project was the first global observation system for the subsurface ocean and is

one of the best sources for in situ temperature measurements. The unprecedented spatial-temporal sampling and coverage of the Argo project has created a unique and massive dataset of subsurface temperature records that can be used to study the distribution and variability of thermocline depth. In this analysis, the Argo floats data were provided by the Coriolis Global Data Acquisition Center of France through their website: www.coriolis.eu.org. The quality control and processing of Argo data are conducted automatically by the Argo data center (Wong et al., 2003; Bohme and Send, 2005; Owens and Wong, 2009).

Based on Chaigneau et al. (2011), we applied a data filtering process to meet the following criteria:

1. The shallowest data point of the Argo profile is located between the sea surface and 10 dbar, and the deepest data point is deeper than 1000 dbar;

2. The depth interval between two consecutive data points must not exceed the following ranges:

a. The intervals between 0–100 dbar must not exceed 25 dbar;

b. The intervals between 100–300 dbar must not exceed 50 dbar;

c. The intervals between 300–1000 dbar must not exceed 100 dbar;

3. Each Argo profile has at least 30 valid data points above 1000 dbar.



After preprocessing and filtering the data, a total of 1,881,318 profiles were retained for analysis.

## 3 Methods

### 3.1 Eddy identification

For eddy identification, we established a four-step scheme based on the earlier works by Chelton et al. (2011), Mason et al. (2014), Liu et al. (2016), and Tian et al. (2020). First, a high-pass filter using a Gaussian filter with a zonal/meridional radius of 10 °/5 ° was applied to the global SLA data before seed points (i.e., SLA extrema) were determined. Second, the global SLA fields were divided into 8 × 5 blocks with a zonal/meridional spacing of 45 °/36 °. Third, SLA contours were computed using a 100 0.25-cm interval, followed by extraction of eddy boundaries (as detailed below). Finally, all blocks were merged seamlessly into a global map and duplicated eddies were eliminated.

To identify cyclonic (or anticyclonic) eddies, an SLA contour was searched from the minimum (maximum) value up (down) to zero to see if it satisfied the following criteria:

1. Contained no more than one "seed" (local maximum/minimum);

2. Nine pixels ≤ eddy size ≤ 2,000 pixels (ensures mesoscale);

3. The eddy area within the equivalent circle exceeded 55%;

4. Eddy amplitude ≥ 2.0 cm.

An eddy was confirmed when it passed the above criteria, and the outermost closed SLA contour was then used as the effective perimeter of the eddy.

### 3.2 Eddy matching using Argo data

Based on the positions and times of eddies and Argo floats, the Argo profiles were classified into three categories depending on whether the floats were inside the effective boundaries of the altimeter recognized anticyclonic or cyclonic eddies, or outside the eddies. The numbers of profiles that fell inside AE or CE in this study were 258,652 and 260,331, respectively, and 1,362,335 profiles were outside eddies. Figure 1 shows the geographical distribution of Argo profiles. The profiles inside AE 115 and CE (Figure 1b) were used to analyze the effect of eddies on the thermocline, while profiles outside eddies (Figure 1c) were used to calculate the background field. Using these methods, combining satellite data and Argo data, the impact of eddies on the thermocline was investigated.

### 3.3 Thermocline detection

The thermocline layer is characterized by a steep vertical temperature gradient. It is often defined by the depth of a 120 'representative' isotherm (e.g. the depth of the 20°C or 14°C isotherm) within the thermocline layer because that isotherm is generally located near the center of the main thermocline (Kessler, 1990). It is the most convenient and simple method to identify the thermocline depth. However, the 14 ℃ isotherm is much too deep to represent the warm pool thermocline (Wang





et al., 2000) and the 20 ℃ isotherm may fail to reflect realistic long-term trends of thermocline strength (Yang et al., 2009). In this study, the depth of thermocline (DTC) was taken as the depth of the maximum vertical temperature gradient based on

individual Argo profiles, which has been shown to effectively reflect actual DTC values (Yang et al., 2009). The thermocline intensity $T_z$ is the magnitude of the sea temperature change over the vertical distance, calculated following Eq. (1):

$$T_z = \frac{\Delta T}{\Delta Z} = \frac{T_{up} - T_{low}}{Z_{up} - Z_{low}},$$ (1)

where $T_{up}$ and $T_{low}$ are the temperatures at depth of $Z_{up}$ and $Z_{low}$, respectively. The maximum thermocline intensity is obtained by comparing the changes in temperature over all vertical ranges, and the depth corresponding to the maximum

change per unit distance is the DTC. In addition, the near-surface layer depth was set to -20 m to reduce the effect of noise near the ocean surface.

### 3.4 Normalization analysis method

In order to reflect the thermocline depth distribution inside eddies more clearly and intuitively, we use transformed the geographic distributions of thermocline depths into the eddy coordinate system. The normalization method used to acquire the

normalized eddy contained four steps:

Step 1: Search for the nearest eddy around the float and calculate the distance between the float and eddy. The normalized distance is calculated as the absolute distance divided by the effective radius of the eddy. Only normalized radii less than 2 are selected.

Step 2: Establish a grid with the eddy center as the center. The depth of the thermocline calculated at the float is then located

in the eddy coordinate system with the eddy center as the center.

Step 3: Reconstruct the eddy field using the floats' time and polarity. Different normalized eddy fields can be constructed by averaging all the AE, CE, DTC, or eddy-induced DTC (EDTC) values, making sure they have the same center as the climatological normalized AE or CE fields.

Step 4: Use Kriging interpolation and Gaussian filtering to get the regularized and smoothed normalization results.

## 4 Results

### 4.1 Global temporal and spatial distributions of DTC

The annual and monthly average shifts of the DTC inside AE and CE in the Northern Hemisphere (NH) and Southern Hemisphere (SH) are shown in Figure 2. The DTC exhibited strong annual, semiannual and seasonal cycles. The DTC inside CE were shallower than inside AE during the 17-year time series in both the NH and SH, and there was a larger difference

between the DTC inside AE and CE is in the first half of the year in the NH, and the opposite was true in the SH (Fig. 2a and 2b). And spring is the strongest season, which is March, April, May (MAM) in the NH and September, October, November



(SON) in the SH. We further analyzed the monthly changes and saw that, while AE deepened the DTC and CE thinned it, the difference between the DTC in AE and CE varied by month, with more pronounced differences in February, March, and April (FMA) in the NH and in August, September, and October (ASO) in the SH. The magnitudes of the differences were largest in

March and September in the Northern and Southern Hemispheres, respectively. These differences were likely related to the dynamics of the mixed layer, which is very thin or nonexistent in summer and autumn and thick, sometimes hundreds of meters thick, in winter and spring due to both wind and surface buoyancy forcing.

The global monthly geographical distribution of the DTC inside AE and CE was obtained and is shown in Figure 3. By and large, the spatial distribution of DTC revealed many remarkable features including significant monthly variation, regional

characteristics, and differences between hemispheres. In the NH, the thermocline is deeper in FMA, with the deepest DTC values concentrated in the Northwest Pacific (NWP) and the Northern Atlantic (NA). The deepest DTC were more than 300 m deep for both AE and CE. However, starting in May the DTC moved shallower, reaching their most shallow depths in August, sometimes as high as 30 m from the surface. This phenomenon was the opposite in the SH, where the DTC was very shallow between 30 °S–40 °S in December, January, and February, gradually becoming deeper than 200 m in ASO. Notably,

the thermocline was comparatively deeper throughout the year in the Antarctic Circumpolar Current, especially in winter and spring.

Looking at the entire year in oceans worldwide, the DTC inside AE were deeper than inside CE, and this difference was most significant in the NWP and NA from February to April and in the North Indian Ocean and South Pacific from September to October. These regions exhibited strong eddy activity and were enriched with middle-aged and long-lived eddies (Chen and

Han, 2019). The stability of eddies and strength of eddy activity both influenced the depth of the thermocline, with more stable and stronger eddies leading to deeper DTC.

**4.2 Eddy-induced depth of thermocline**

In order to quantitatively study the effects of AE and CE on vertical shifts of thermoclines, the eddy-induced depth of thermocline (EDTC) values were obtained by removing the background field from the thermocline. The background DTC was

calculated from Argo profiles outside of the altimeter identified eddies. The global distribution of EDTC and the average EDTC for latitudinal zones are presented in Figure 4. Globally, eddies generally induced DTC shifts of between 5 to 15 m deeper. There was a distinct spatial distribution for the EDTC shifts, with larger DTC shifts concentrated in the midlatitudes, meanwhile much larger DTC shifts appeared along the western boundaries of strong current systems such as the Kuroshio extension (KE), Gulf Stream, North Atlantic warm current, East Austrian warm current, Brazil warm current, and Agulas

current, with peak shifts of more than 40 m. These regions of large EDTC were also accompanied by large eddy amplitudes (Chelton et al. 2011), which further demonstrates that stronger eddies have larger influences on DTC. Figure 4c and 4d exhibit distinct latitudinal patterns, where the EDTC maxima were found in the subtropics and in the midlatitudes, with a maximum of ~25 m. On the contrary, the EDTC was relatively small in lower and higher latitude regions, only about 5 m. The above analysis clearly demonstrated that eddies can induce vertical shifts in DTC: with warm eddies, because of warm water sinking,



drive thermoclines to sink, and cold eddies, because of cold water rising, drive thermoclines to rise. The influence of eddies on the depth of thermoclines originates from the mechanisms that transport seawater within the eddy. The divergence of CE causes the surface seawater to disperse outwards and deep cold water is transferred upward to replace the surface water, exerting an uplifting force on the thermocline. The AE, by contrast, causes the surface water to converge inward, and the thermocline is forced lower.

Furthermore, we examined the ETDC distribution during the seasons when eddies had the greatest effect (MAM in the NH and SON in the SH; Fig. 2). Both AE and CE had a strong influence areas in the KE and Gulf Stream during MAM (~70m) with a ± 10 m impact in most regions of the global ocean. However, there were some regions in the SH that exhibited the opposite effect, which may have been caused by movements of other large water masses (Fig. 5a and 5b). It was also found that a systematic phase reversal existed during SON. During this period, the extremes of the NH disappeared while the SH

showed marked increases in the Brazilian coastal current, East Australian current, and Agulhas current (~60m), but these trends were weaker in CE (Figure 5c and 5d).

Since large EDTC values appeared in regions with strong eddy activity, we analyzed the correlations between the eddy properties and EDTC values. Figure 6 shows EDTC had an almost linear relationship with eddy radius and amplitude after least square fitting, suggesting that strong activity in eddies tends to induce a larger DTC shift. In addition, the relationship

between eddy amplitude and EDTC was stronger, that is, eddy with larger amplitude had a larger effect on EDTC.

**4.3 Thermocline distributions in eddy coordinate systems**

The influences of earth coordinates on eddy characteristics are remarkable (Chen et al., 2019). Therefore, to reveal the characteristics of the inner thermocline depth changes of eddies, we carried out a circular normalization composite analysis to transform the global thermocline inside eddies into the eddy coordinate system. Figure 7 shows the mean DTC and EDTC

changes as revealed by the normalized AE and CE, respectively. The DTC inside AE had a positive maximum of around 200 m in the eddy center (Fig. 7a). However, the CE exhibited a concave valley structure with the lowest value being about 120 m at the eddy center. The farther we move away from the eddy center the higher the DTC became, which means the DTC inside CE were encircled by deeper thermoclines (Fig. 7b). When we move the background field, a monopole structure appears. At the normalized eddy center, AE increased the DTC by ~30 m (Fig. 7c). The same behavior was observed for the normalized

CE (Fig. 7d), but in the opposite direction, with a decrease in DTC by ~20 m.

The seasonal changes in the NH are presented in Figure 8. These demonstrate that the centers of AE have a high DTC values throughout the year (Fig. 8, left panels), and were higher than the DTC at the surrounding edges of eddies. In contrast, the centers of CE had relatively low DTC values throughout the year (Fig. 8, right panels), and there was a ring of higher values around the perimeter of its effective radius. The seasonal normalized AE revealed positive changes of EDTC in every season

(Fig. 9, left panels), while the seasonal normalized CE exerted generally negative changes (Fig. 9, right panels). In the seasonal mean normalized values of both CE and AE, the changes in EDTC in December, January, and February (DJF) and March, April, and May (MAM) were much larger than those in June, July, and August (JJA) and September, October, and November (SON).



The mean changes in EDTC associated with the seasonal normalized eddies in the NH are presented in Table 1. The effect that
eddies had on the displacement of the thermocline was the most pronounced in MAM. The climatological mean DTC during
MAM was ~139 m. At the eddy center, the normalized CE decreased the DTC thickness by ~19m, while the normalized AE
increased it by ~22m. In other words, the climatological mean EDTC was thinned by approximately 13.7% at the center of
normalized CE, while it was thickened by 15.8% at the center of normalized AE. In JJA, the effect of eddies on the rise and
fall of the thermocline was weakest, with falls in AE averaging only 7 m and rises in CE averaging only 8 m.
The same calculation was made for the eddies in the SH. The results of DTC normalization and EDTC normalization are
shown in Figures 10 and 11. The effect of eddies in the SH was smaller than in the NH, but AE and CE can still be seen to
deepen and uplift the DTC, especially in JJA and SON. The highest DTC values were observed at the center of the AE and the
ring of high values around the CE were more pronounced during these months. After removing the background field, the
EDTC at the eddy center in JJA and SON was more obvious, and the deepening effect of the AE was stronger than the uplifting
effect of the CE. Table 2 shows that the extent of the change in EDTC associated with the seasonal normalized AE was largest
in SON (~13.7%), and the smallest was 5.8% in MAM. In contrast, the largest change in EDTC associated with the seasonal
mean normalized CE was 10.5% in SON, and the smallest was only 1.2% in MAM.

## 5 Summary and Conclusions

Using a global, 17-year spanning, Argo dataset in combination with satellite altimetry, we observed how the vertical depth of
thermocline (DTC) was affected by anticyclonic eddies (AE) and cyclonic eddies (CE). The major findings of the study are
summarized as follows.

First, thermoclines were relatively deeper and shallower in AE and CE, respectively. In the CE, the diverging surface water
caused deep, cold water masses to move upward which uplifted the thermocline, and in the AE the convergence of surface
water had the opposite effect, pushing the thermocline deeper. The DTC exhibited obvious annual and semiannual cycles. It
gradually moved shallower from February to July and deeper from July to the following January in the NH, with the opposite
being true in the SH. Such cycles are closely related to the underlying dynamic mechanisms in the ocean, such as turbulent
mixing, upwelling and downwelling (Ekman pumping), fluid trapping, and particle entrainment, as well as air-sea interactions.
Second, the global distribution of the changes in EDTC showed remarkable latitudinal trends. Globally, eddies can induce
shifts in the DTC with magnitudes of around 5 m to 15 m. However, this effect is more pronounced along the western
boundaries of strong current systems, with peak shifts of more than 40 m. To generalize, eddy-induced shifts in DTC appeared
to have strong linear relationships with eddy radius and amplitude, which means that strong, high intensity eddies induce a
greater DTC displacement. This correlation meant that the geographical distribution of change in DTC of eddies was similar
to those of eddy properties (especially eddy amplitude).

Third, the normalized analysis indicated that there were monopole (ring) distributions of shifts in DTC inside AE and CE.
From the global normalization for the whole year, we saw a shift of about 30 m at the center of AE, while in CE the shift was

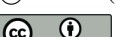

only 20 m. Obviously, the effects of AE and CE on DTC was asymmetrical. One mechanism that may explain this asymmetry is the differential current shears in the thermoclines. That is, because of the stratified structure of sea water and the baroclinic instability caused by the movement of water masses, a downward force is produced in the thermocline, which would favor moving water masses deeper. Another potential factor would stem from our recent research findings that there is a higher

proportion of abnormal CE, compared to AE. In other words, a (an) cyclonic (anticyclonic) eddy is usually associated with a cold (warm) core in its vertical structure. However, there are some cyclonic (anticyclonic) eddies with warm (cold) cores which we called them abnormal eddies. There is a larger proportion of CE with a warm temperature anomaly in its vertical structure based on Argo profiles but is judged as a cyclonic eddy by the altimeter. This kind of misjudgment may lead to a weaker influence of CE on the displacement of thermocline depth.

In conclusion, this study systematically and quantitatively analyzed and estimated the role of mesoscale eddies on thermocline depth globally. The results showed that the DTC was deepened or uplifted depending on characteristics associated with time, space, and the polarity of eddies. Further research is needed to explain the mechanisms behind these systematic differences and associations to further deepen our understanding of this phenomenon.

**Data availability**

The data used to support the findings of this study are available as follows. The altimeter all-sat merged sea level anomaly (SLA) data used in this study are distributed by Archiving, Validation, and Interpretation of Satellite Oceanographic (AVISO; https://www.aviso.altimetry.fr/index.php?id=3122). The eddy identification dataset can be obtained from http://coadc.out.edu.cn/tfl/, and Argo data can be obtained from www.coriolis.eu.org.

**Author Contributions**

Xiaoyan Chen performed methodology, software, validation, data curation, visualization, and writing of the original draft. Ge Chen contributed to conceptualization, investigation, writing review and editing, and supervision.

**Acknowledgements**

This research was jointly supported by the Ministry of Science and Technology of the People's Republic of China under grant no. 2016YFC1401008, and the Qingdao National Laboratory for Marine Science and Technology under grant no.
2018SDKJ0102.

**Competing interests**

The authors declare that they have no conflict of interest.





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





Figure 1: Heatmaps of the geographical distributions of (a) all Argo profiles, (b) Argo profiles inside eddies, and (c) Argo profiles outside eddies on 5 °× 5 ° grids, as determined by the altimeter data, during 01/01/2002–01/12/2019.





**Figure 2: Monthly time series of DTC inside AE (red lines) and CE (blue lines) in the (a) Northern and (b) Southern Hemispheres during 2002-2019. (c) and (d) show averaged monthly DTC for the Northern and Southern Hemispheres, respectively.**



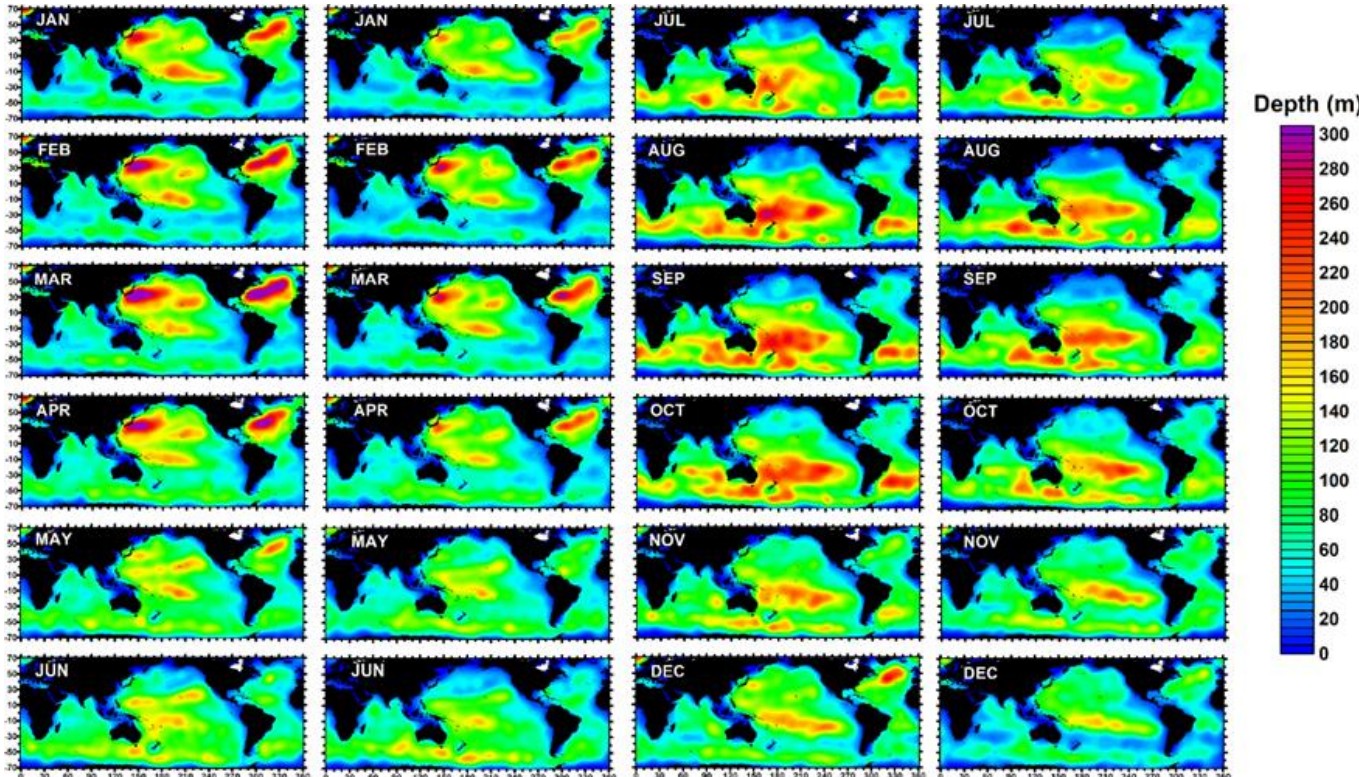


**Figure 3: Map of monthly means DTC of AE (first and third columns) and CE (second and forth columns) from January to December (2 ° × 2 ° resolution).**


**Figure 4: (a-b)** Maps of eddy-induced depth of thermocline for the global ocean for AE (a) and CE (b). **(c-d)** Averages of EDTC for latitudinal zones from (a-b). A two-dimensional smoothing was applied to (a-b) with a moving 3 ×3 window and a three point smoothing was applied to (c-d).

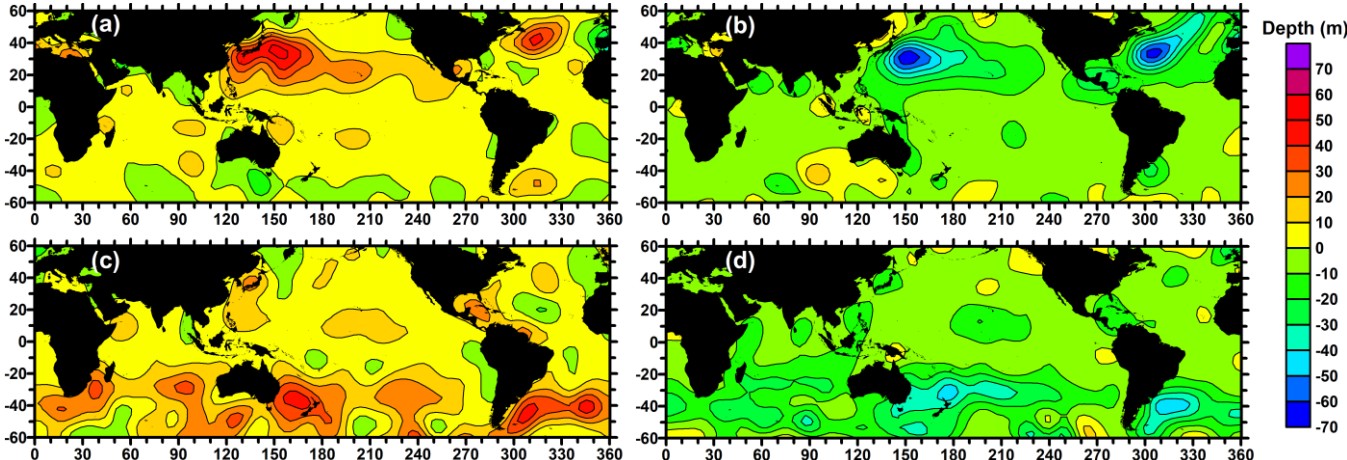

**Figure 5: (a-b)** Map of global eddy-induced depth of thermoclines in MAM for AE (a) and CE (b) and in SON for AE (c) and CE (d). Two-dimensional smoothing was applied to all maps with a moving window of 3 ×3.






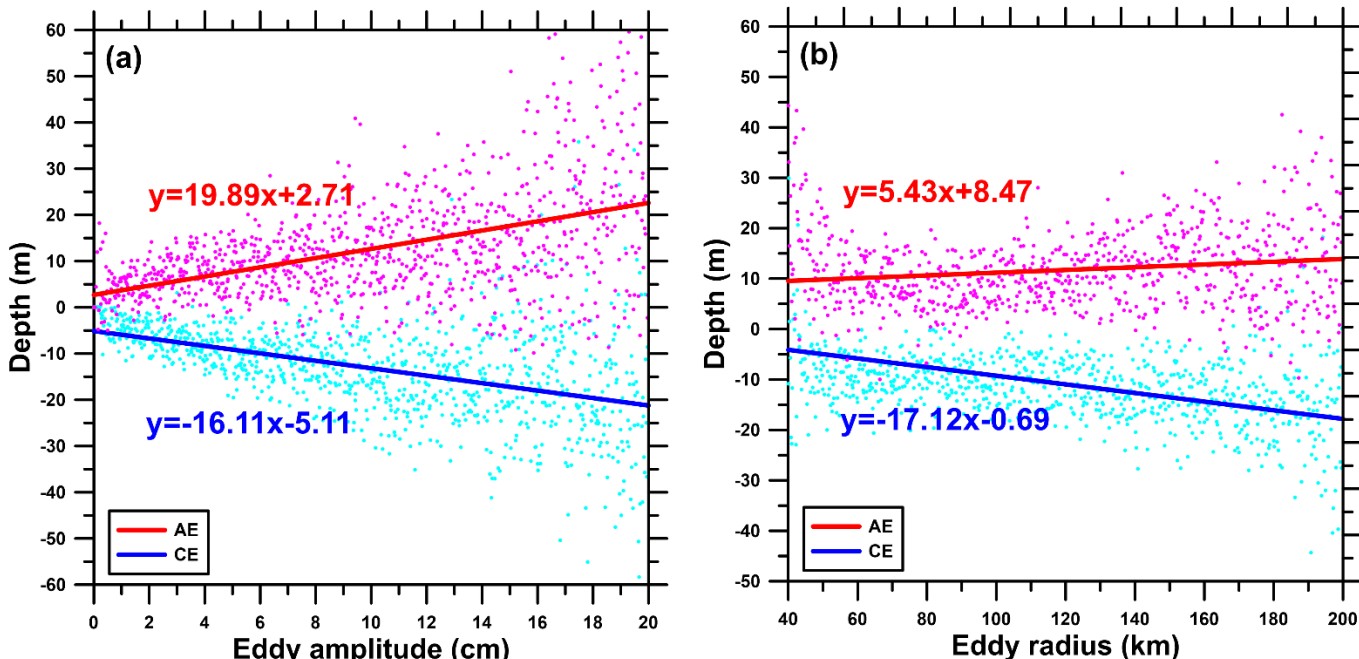

**Figure 6: Eddy amplitude (a) and eddy radius (b) plotted against eddy-induced thermocline depth for AE (red lines) and CE (blue lines). Light blue and magenta dots represent the average ETDC at the corresponding amplitude or radius of AE and CE, respectively. Red and blue lines are the least square fitting results of corresponding points.**



Figure 7: Distributions of DTC (left panels, a-b) and ETDC (right panels, c-d) inside eddies after circular normalization. The top panels are for AE and bottom panels are for CE.





**Figure 8: Circular normalized composite distributions of DTC in AE (left panels) and CE (right panels) in the Northern Hemisphere. From top to bottom, rows show DJF, MAM, JJA, and SON.**





**Figure 9:** Circular normalized composite distributions of EDTC in AE (left panels) and CE (right panels) the Northern Hemisphere. From top to bottom, rows show DJF, MAM, JJA, and SON.



| Month | Background DTC (m) | Eddy type | EDTC (m) | Anomaly percent (%) |
|---|---|---|---|---|
| DJF | 135 | AE | 18 | 13.3 |
|  |  | CE | -14 | 10.4 |
| MAM | 139 | AE | 22 | 15.8 |
|  |  | CE | -19 | 13.7 |
| JJA | 70 | AE | 8 | 11.4 |
|  |  | CE | -9 | 12.9 |
| SON | 77 | AE | 7 | 9.1 |
|  |  | CE | -8 | 10.4 |

**Table 1: Normalized mean-EDTC during the four seasons in the Northern Hemisphere**

*Note.* DJF = December, January, and February; MAM = March, April, and May; JJA = June, July, and August; SON =
September, October, and November.







Figure 10: Same as Figure 8 but in the Southern Hemisphere. Circular normalized composite distributions of DTC in AE (left panels) and CE (right panels) in the Southern Hemisphere. From top to bottom, rows show DJF, MAM, JJA, and SON.







| Month | Background DTC (m) | Eddy type | EDTC (m) | Anomaly percent (%) |
|---|---|---|---|---|
| DJF | 93 | AE | 6 | 6.5 |
| | | CE | -5 | 5.4 |
| MAM | 86 | AE | 5 | 5.8 |
| | | CE | -1 | 1.2 |
| JJA | 144 | AE | 13 | 9.0 |
| | | CE | -8 | 5.6 |
| SON | 153 | AE | 21 | 13.7 |
| | | CE | -16 | 10.5 |

**Table 2. Normalized mean-EDTC in four seasons in the Southern Hemisphere**