# Peer review of "Effect of mesoscale eddy on thermocline depth over the global ocean: deepen and uplift"

_Ocean Science, 2020_

## Referee Comment (RC1) · Anonymous Referee #1 · 18 Aug 2020

Based on Argo dataset and satellite altimetry, this study examined effects of mesoscale eddy on thermocline depth over the global ocean. The authors reached some interesting conclusions. The manuscript is clearly written and logically organized. I suggest to accept this manuscript after addressing the following concerns.

1. Lines 30-45. "The role eddies play in large scale ocean circulation and transport systems...". "...anticyclonic eddies cause thermoclines to deepen..." Mesoscale eddies significantly modulate the thermocline depth and thus could induce larger eddy transport in the thermocline. Please read Chen et al. (2012) for details.

Chen, G., J. Gan, Q. Xie, X. Chu, D. Wang, and Y. Hou (2012), Eddy heat and salt transports in the South China Sea and their seasonal modulations, Journal of Geophysical Research, 117(C5), C05021, doi:10.1029/2011JC007724.

[Figure]

2. When you reviewed effects of mesoscale eddies on the thermocline, I suggest to:

(1) quantify eddy-induced thermocline anomalies

(2) emphasize the spatial discrepancy of eddy-induced thermocline anomalies.

To this end, you may want to read, for example:

Chen, G., Y. Hou, Q. Zhang and X. Chu (2010), The eddy pair off eastern Vietnam: Interannual variability and impact on thermohaline structure, Continental Shelf Research, 30, 715-723.

Chu, X., H. Xue, Y. Qi, G. Chen, Q. Mao, D. Wang, F. Chai (2014). An exceptional anticyclonic eddy in the South China Sea in 2010, Journal of Geophysical Research, doi: 10.1002/2013JC009314.

Lasitha Perera, G., G. Chen, M. J. McPhaden, T. Priyadarshana, Ke. Huang and D. Wang (2019). Meridional and Zonal Eddy-Induced Heat and Salt Transport in the Bay of Bengal and Their Seasonal Modulation. Journal of Geophysical Research: Oceans, 124. https://doi.org/10.1029/2019JC015124.

Chu, X., G. Chen, and Y. Qi (2020). Periodic mesoscale eddies in the South China Sea. Journal of Geophysical Research: Oceans, https://doi.org/10.1029/2019JC015139.

3. You define the thermocline intensity $T_z$ in Section 3.3. Have you shown any results with this parameter?

4. Lines 148-150. Please try to quantify the results. The temperature shown in Figure 2 is related to not only the eddies but also the background temperature. Could you show the temperature cycle with no eddy for comparison? What does the black line in Fig. 2c and 2d represent? Please add some explanation about the black line in Figure 2 caption.

5. Are the results shown in Figures 3-5 affected by the uneven distribution of Argo profiles (shown in Figure 1)?

6. In addition to eddy amplitude and eddy radius shown in Figure 6, you may want to show the relationship between eddy-induced thermocline depth and other eddy parameters, e.g., the mean eddy kinetic energy, the mean vorticity, and so on.

7. Line 46: "layer has little to no effect on the exchanges"?

8. Line 198: "had an almost linear relationship with". It seems that the relationship shown in Figure 6 is not linear when the amplitude is larger or the radius is smaller. You may want to try the least square fitting with a curve instead of the straight line.
* * *

---

## Referee Comment (RC2) · Anonymous Referee #2 · 19 Aug 2020

In this manuscript the authors examine the "effect" of mesoscale eddies on thermocline depths over the global ocean by using altimeter sea surface height data and Argo profiling float data.

In mid-latitudes, both seasonal and permanent thermoclines are likely to be detected, but the authors did not distinguish these two in their analysis. Probably they mainly detected the permanent thermocline in cold seasons and the seasonal thermocline in warm seasons. The vertical shift of the permanent thermocline is the eddies' structure itself, and not their "effect". The anticyclonic and cyclonic eddies might have some effect on the formation of seasonal thermocline, but such detailed analysis has not been performed in this manuscript. What about higher latitudes, especially in the Southern Ocean and subarctic North Pacific where the ocean is stratified by salinity? These regions probably lack thermocline in cold seasons at all depths. A global analysis without considering such regional differences seems meaningless.

Due to poor analysis and numerous descriptions without scientific basis, I cannot recommend the publication of this manuscript in Ocean Science.

———————————————

---

## Referee Comment (RC3) · Anonymous Referee #3 · 19 Aug 2020

In this manuscript, the authors show the spatial and seasonal changes in mixed layer thermocline depths both in anticyclonic and cyclonic eddies with Argo profiles and satellite SSH. Though the analysis was interesting, I feel additional analysis and discussions were needed to clarify the advantage of this study.

In this study, the authors focused on the changes in the thermocline depths. As the authors did not distinguish the seasonal thermoclines from permanent ones, almost of all the changes probably represent the seasonal thermocline changes, which are generally located just below mixed layers. So, the thermocline depth changes in this study can be very similar to the mixed layer depths. Actually, seasonal changes (Figs. 2c and d), spatial distributions (Figs. 4 and 5), and the relationships with the amplitudes (Fig. 6) were similar to the ones in Gaube et al. 2018, which was cited by the authors in this

manuscript. The detailed comparison with Gaube et al. 2018 would be needed to clarify the difference and similarity of the changes in MLD and thermocline depths because the authors seemed to emphasize changes in the thermocline not in the mixed layers (around L. 45-54). It may be useful to show gradients of thermocline to clarify differences from Gaube et al. 2018. I think the temperature and salinity changes around eddy thermocline depths were focused on clarifying the eddy roles in heat/freshwater transports by previous studies. If the authors focus on this theme, comparison with the results Sun et al. (2019; doi:10.1038/s41598-018-38069-2) may be useful. They estimate the trapping and stirring effects in heat transport. The former may be related to the thermocline (gradient/depth) changes in the eddy core, and the latter may be related to the changes with the edge (Figs. 7-9 in this study).

At least, the analysis to clarify the differences of changes in thermocline from the ones in MLDs is needed before publication. The main results may be fully rewritten based on the new analysis. So I would like to recommend the authors to withdraw at this time and to resubmit.

---

## Author Comment (AC1) · 24 Aug 2020

Dear referee: We would like to express our sincere appreciation for your careful reading and invaluable comments. In response to your suggestions, I have some of my point of views as follows. Thermocline and geostrophic vorticity are closely related to each other and mesoscale eddy plays an important role in modifying the thermocline structure for their ability of transporting heat, salt and other chemical substances, and changing dynamic conditions of the ocean. The effect of mesoscale eddies on the thermocline that we emphasize is the uplifting and deepening of eddies on the thermocline. It is closely related to eddies' structure. The horizontal scale that a single eddy can affect is on the order of hundreds of kilometers, and the vertical scale is thousands of kilometers. Oceanic eddies are omnipresent in the ocean, accompanied by the slowly

migration process of eddies (∼10km/day), the background stratification of the ocean is re-adjusted. Our job is to select a standard stratification to measure the influence of this adjustment. It is undeniable that the mesoscale eddy has a modulating effect on both the thermocline and the mixed layer. But thermocline defined by maximum temperature gradient is a more stable stratification structure, so the displacement of the thermocline is less interfered by other factors such as air-sea interface interaction while is mainly captured by mesoscale dynamics. There are many corresponding researches about the effect of mesoscale eddy on the structure of thermocline. Except as mentioned in the manuscript, Liu et al. (2001) pointed out that the thermocline changes when an eddy passes by. Cessi and Fantini (2004) indicated that baroclinic eddies on scales from 50 to 100 km could maintain a thermocline against diapycnal diffusion. Zhang et al. (2010) found that the eddy had a great influence on the thermohaline structure pattern in the local upper ocean, that is, when the eddy was strong, the thermocline depth shoaled greatly, and the subsurface high salinity water decreased largely. However, most of the existing related researches are based on a specific sea area. With the abundance of Argo data, we expand the research area to the entire global sea area, so that the thermocline displacement caused by eddies can be seen from a global perspective and long-term series. The spatial distribution feature of eddy-induced thermocline displacement is also one of the highlights of this article. We will be more inclined to dig out information and characteristics based on the ocean big data and display them quantitatively to reveal an actual ocean phenomenon. In addition, we obtain the position of the thermocline by calculating the maximum gradient of the temperature profile data obtained by each Argo float, as shown in Figure 1. When studying the displacement of the thermocline, the conclusion that the displacement of the thermocline caused by the eddy is based on the comparison between the thermocline depth of the inner eddy and the depth of the outer eddy. In other words, the changes in the thermocline itself are considered when calculating anomalies and it is reasonable to use the thermocline stratification with the fastest sea temperature change as a representative. What's more, you mentioned that we did not consider regional differences
in our research. In fact, you can see obvious regional characteristics in Figure 4-5 in the manuscript that remarkable thermocline displacements appeared along the western boundaries of strong current systems such as the Kuroshio extension, Gulf Stream, North Atlantic warm current, East Austrian warm current, Brazil warm current, Agulas current. But we indeed did not expand the areas with these characteristics for further research which is also worth doing later. The effect of eddy on the thermocline we have calculated is based on the identified eddy data set. Therefore, the study of the spatial distribution of the eddy-induced thermocline displacement could be restricted by the spatial distribution of eddies. And for the regional where lack thermocline, the effect of eddy on the thermocline is naturally small or even non-existent. However, many recent studies have shown that eddies also have uplifting and sinking effects on the halocline, which further shows that the phenomenon of ocean stratification displacement caused by such eddies is ubiquitous. The effects of eddies on halocline and pycnocline, and the differences in these effects are also our future research directions.

[Figure]

**Fig. 1.** Procedure for determining the thermocline depth a. Argo temperature profile; b. vertical temperature gradient of the Argo temperature profile.

---

## Author Comment (AC2) · 24 Aug 2020

Dear referee: Thank you for your nice comments concerning our manuscript. We would like to express our sincere appreciation for your careful reading and invaluable comments. Gaube et al. 2018 (here after Ga2018) has done a great job in mixed layer depth (MLD) displacement (deepen and shoal) modulated by mesoscale eddies. There are some similarities with the results in Ga2018, but there are indeed many differences that need to be elaborated for you. Firstly, the mix layer is located on the surface of the ocean, it is a manifestation of the vigorous turbulent mixing process which is active in the upper ocean. The transfer of mass, momentum, and energy across the mixed layer provides the source of almost all oceanic motions. Therefore, in addition to the mesoscale eddy, the displacement of the mixed layer is more susceptible to the

influence of air-sea interaction (e.g. sea surface wind field or air pressure field abnormal). Ga2018 studied the modulation effect of the eddy on the MLD, which is actually a comprehensive assessment of the MLD migrate that naturally includes the influence of not only eddies but also the air-sea interface interaction, and has not been eliminated. While in our research, we obtain the position of the thermocline by calculating the maximum gradient of the temperature profile data obtained by each Argo float, as shown in Supplementary Figure 1. In fact, we believe that eddies will have an uplift and deepen effect on both the thermocline and the mixed layer. However, what we want to highlight is that the thermocline defined by maximum temperature gradient is a more stable stratification structure, so the displacement of the thermocline is less interfered by other factors while is mainly captured by mesoscale dynamics. Thus, our research on the thermocline depth displacement is closer to the evaluation of deepen and uplift on the ocean stratification by the mesoscale eddies itself. Secondly, comparing with the study of Ga2018, it is found that the seasons of the peaks of eddy-induced depth displacement are different. In their research, the largest displacement is shown in winter, while our research found that the displacement is most significant in spring. This seasonal delayed response indicates the difference in the response of the thermocline and the mixed layer, which highlights our quantitative assessment of eddies' effect on thermocline. What's more, Ga2018 proposed that the magnitude of eddy-induced MLD anomalies is largest during the winter in regions of large eddy amplitude (figure 1 in Ga2018), check this figure we can easily find out only a few grid points near the +50m in AE and -50m in CE at some major strong current regions, however, the values are up to +60m in AE and -70m in CE during the MAM and SON at same regions in our statistics. Meanwhile, the coverage of the high value is large than Ga2018 remarkably (see figure 5 in our manuscript). The magnitude and coverage of the depth anomaly are representing the effect of eddies in ocean stratification. Argo profile data were used in both studies, and the SSH-based automated eddy identification method (Chelton et al. 2011) were used in AVISO multi-altimeter merged SLA data to construct the mesoscale eddy datasets, but different effects were assessed. Last but not least,

the letter by Gaube focused on revealing this global feature, but the quantitative evaluation is not detailed enough. Our research shows richer data information and more comprehensive characteristics. For example, we quantified the eddy-induced thermocline displacement in different seasons, and transformed the geographic distributions of thermocline depths into the eddy coordinate system to visualize the distribution of thermocline displacement in the eddy field. What's more, in addition to the eddy amplitude, we give the relationship between eddy radius, geostrophic velocity, eddy kinetic energy and thermocline displacement, and point out the relationship between the eddy lifetime and its displacement distribution which is more comprehensively reveal the relationship between eddy properties and thermocline displacement. In addition, although we all pointed out that the deepening effect of AE is stronger than uplifting of CE is due to the differential current shears in the thermoclines, we pointed out a new explanation for this phenomenon that stem from our recent research findings that there is a higher proportion of abnormal CE, compared to AE. And this unbalanced abnormal eddy ratio may lead to a weaker influence of CE on the displacement of thermocline depth. This has not been pointed out before and could provide a reference for subsequent research.

[Figure]

**Fig. 1.** Procedure for determining the thermocline depth a. Argo temperature profile; b. vertical temperature gradient of the Argo temperature profile.

---

## Author Comment (AC3) · 24 Aug 2020

Dear referee:

We feel great thanks for your professional review work on our manuscript. All of these comments have contributed a lot to improve the quality of our article. And the references tou recommend have been carefully read and are of great help that we will make a careful citation in the final article. According with your advice, we amended the relevant part in manuscript. Some of your questions were answered below.

Q1: You define the thermocline intensity Tz in Section 3.3. Have you shown any results with this parameter?

A: We obtain the position of the thermocline by calculating the maximum gradient of

the temperature profile data obtained by each Argo float. A schematic diagram of the calculation process of a profile is shown in Supplementary Figure 1. Fig. 1 (a) is a temperature profile measured by Argo and interpolated from 0 to 1000m. Fig. 1 (b) is the result of the gradient profile calculated by the corresponding Fig. 1 (a). It can be seen that there is a position with the fastest temperature change at about 98m which is the position of the thermocline at the corresponding coordinate point obtained from this profile data.

Q2: Lines 148-150. Please try to quantify the results. The temperature shown in Figure 2 is related to not only the eddies but also the background temperature. Could you show the temperature cycle with no eddy for comparison? What does the black line in Fig. 2c and 2d represent? Please add some explanation about the black line in Figure 2 caption.

A: Figure 2 shows the annual and monthly changes of the thermocline depth inside AE, CE and outside eddies in the northern and southern hemispheres from 2002 to 2019. (a) And (b) correspond to the annual change, (c) and (d) correspond to the monthly change. We have added a black curve to indicate the depth of the thermocline which is outside eddies, so as to compare with the depth of the thermocline inside the eddies (that is to say, no eddy for comparison). We also added a black curve in Figure 2 (a-b) and corrected the figure caption, see Supplementary Figure 1. This figure can mainly reflect the following information: Almost the entire time series, the depth inside CE is shallower than AE, and exhibited strong annual, semiannual and seasonal cycles. Quantitatively, from the monthly average of the northern hemisphere, AE can cause the thermocline to sink by nearly 40m around March, while CE can cause uplift of approximately 20m in March and April. As the months go by, the difference between the effects of AE and CE gradually decreases, reaching the minimum in August to September, only about less than 5m. In the southern hemisphere, AE can cause a maximum of about 25m of thermocline deepening during September to October, while CE can cause uplift of less than 10m in September. In March to April, the uplifting effect

of CE on the thermocline almost disappeared, and AE can only cause the thermocline displacement of about 2m.

Q3: Are the results shown in Figures 3-5 affected by the uneven distribution of Argo profiles (shown in Figure 1)?

A: We tend to suppose that the results in figures 3-5 are more affected by the property of eddies (such as eddy amplitude, geostrophic velocity, radius, etc.). We added supplementary figure 3 as an illustration. Supplementary Figure 3 shows the displacement in the depth of the thermocline calculated by the Argo profiles that outside the effective boundary of the eddy, but within the boundary corresponding to twice its effective radius. This can be compared with figure 4 (a-b) in the manuscript which is calculated by the Argo profiles that are inside eddies. It can be seen obviously that in areas where Argo is richly distributed, the depth displacement caused by eddy is also very weak. The sampling density of Argo data will affect the accuracy of the results to a certain extent, but this will not be the main effect. And even if the amount of Argo is small in some areas, its density is almost more than 200 for a $5°$ grid, and the thermocline depth can be calibrated relatively accurately. In addition, in areas with large displacement are generally accompanied by strong eddy activities, such as the Kuroshio area and the North Atlantic area. That is, when the eddy was strong, the thermocline depth shifts greatly.

Q4: In addition to eddy amplitude and eddy radius shown in Figure 6, you may want to show the relationship between eddy-induced thermocline depth and other eddy parameters, e.g., the mean eddy kinetic energy, the mean vorticity, and so on. Line 198: "had an almost linear relationship with". It seems that the relationship shown in Figure 6 is not linear when the amplitude is larger or the radius is smaller. You may want to try the least square fitting with a curve instead of the straight line.

A: We made a modification and supplement to Figure 6 in the manuscript that tried the least squares quadratic function fitting, and the fitting effect was indeed better,

with R2 above 0.95. In addition, the two parameters of eddy geostrophic velocity and kinetic energy have been added, as shown in supplement figure 4. After adding these parameters, the relationship between the eddy characteristics and the eddy-induced thermocline displacement can be more fully demonstrated. And we can found the eddy geostrophic velocity and eddy amplitude have the best fitting relationship with a smaller RMSE.

Q5: Line 46: "layer has little to no effect on the exchanges"?

A: This sentence is our statement is not accurate enough. We feel sorry for our carelessness. What this sentence really wants to express is: except during the wintertime convection when the mixed layer deepens significantly, the vertical velocity within the mixed layer that is principally wind-driven does not affect the exchanges between the surface layers and the ocean interior very much. Thank you very much for pointing out, and the manuscript will be revised.

[Figure]

**Fig. 1.** Procedure for determining the thermocline depth a. Argo temperature profile; b. vertical temperature gradient of the Argo temperature profile.

[Figure]

**Fig. 2.** Monthly time series of DTC inside AE (red lines), CE (blue lines) and OE (black lines) in the (a) Northern and (b) Southern Hemispheres. (c) and (d) show averaged monthly DTC for NH and SH.

[Figure]

**Fig. 3.** Maps of outside-eddy-induced depth of thermocline for the global ocean for AE (a) and CE (b).

[Figure]

**Fig. 4.** Eddy amplitude (a), eddy radius (b), eddy u/v speed (c) and eddy kinetic energy (d) plotted against eddy-induced thermocline depth for AE (red lines) and CE (blue lines).

---

## Editor Comment (EC1) · Katsuro Katsumata (Editor) · 10 Sep 2020

The reviewer 1 suggests a major revision. The reviewer points out (comment 4) that Fig.2 includes the effect of "the background temperature". The authors added the black line, showing the results from the analysis performed outside the eddies. The variability of red and blue lines are mostly attributable to the variation of the black line, i.e., non eddy effects. It is suggested to consider the effect of float distribution (comment 5). The authors responded with a figure produced from the floats outside of the eddies. I fail to see how this new figure supports the authors claim that the float distribution will not affect the main effect. The reviewer also suggested to look at the relationship of the thermocline depth with other parameters (comment 6). The authors calculated relationships with geostrophic current and EKE. Under the geostrophic balance, the

eddy amplitude, radius, surface current, and EKE are all correlated and I do not think the new figures add much to the results in the original manuscript.

The reviewer 2 points out problems with the analysis – the effect of thermoclines (permanent and seasonal) contaminates the results. This point is similar to the first point by the reviewer 1. The reviewer also mentions the lack of consideration for salinity. The authors reply is based on a statement that "thermocline defined by maximum temperature gradient is a more stable stratification structure, so the displacement of the thermocline is less interfered by other factors such as air-sea interface interaction while is mainly captured by mesoscale dynamics". This statement is not supported by the analysis but with references which I failed to see a direct link to the reviewer 2's point. The effect on salinity was left for future.

The reviewer 3 again raises the importance to separate the effect of the seasonal thermocline from eddy effects and points out an important reference; Gaube et al. (2018). The reply is based on the same statement that the authors used in their reply to the reviewer 2, and again not supported by data.

Based on these reviews and my own reading, I am not convinced that a revision will improve the manuscript to the point where publication in OS is appropriate (https://www.ocean-science.net/peer_review/review_criteria.html). I discourage submission of a revised manuscript.
* * *